# Brain and Retinal Organoids for Disease Modeling: The Importance of In Vitro Blood–Brain and Retinal Barriers Studies

**DOI:** 10.3390/cells11071120

**Published:** 2022-03-25

**Authors:** Ilenia Martinelli, Seyed Khosrow Tayebati, Daniele Tomassoni, Giulio Nittari, Proshanta Roy, Francesco Amenta

**Affiliations:** 1School of Medicinal and Health Products Sciences, University of Camerino, 62032 Camerino, Italy; khosrow.tayebati@unicam.it (S.K.T.); giulio.nittari@unicam.it (G.N.); francesco.amenta@unicam.it (F.A.); 2School of Biosciences and Veterinary Medicine, University of Camerino, 62032 Camerino, Italy; daniele.tomassoni@unicam.it (D.T.); proshanta.roy@unicam.it (P.R.)

**Keywords:** organoids, brain, retinal, disease modeling, blood–brain barrier, blood–retina barrier

## Abstract

Brain and retinal organoids are functional and dynamic in vitro three-dimensional (3D) structures derived from pluripotent stem cells that spontaneously organize themselves to their in vivo counterparts. Here, we review the main literature data of how these organoids have been developed through different protocols and how they have been technically analyzed. Moreover, this paper reviews recent advances in using organoids to model neurological and retinal diseases, considering their potential for translational applications but also pointing out their limitations. Since the blood–brain barrier (BBB) and blood–retinal barrier (BRB) are understood to play a fundamental role respectively in brain and eye functions, both in health and in disease, we provide an overview of the progress in the development techniques of in vitro models as reliable and predictive screening tools for BBB and BRB-penetrating compounds. Furthermore, we propose potential future directions for brain and retinal organoids, in which dedicated biobanks will represent a novel tool for neuroscience and ophthalmology research.

## 1. Introduction

The term “organoids” has been used to define stem cells with the capacity of self-renewal and self-organization in three-dimensional (3D) structures, containing multiple organ-specific cells, and resembling in vivo conditions [1,2]. The resulting organoids remain committed to their tissue of origin. They express key structural and functional properties of several organs, including the brain [3,4] and retina [5]. Organoid 3D cultures can be generated via a variety of sources, from spheroids derived from adult stem cells (ASCs), embryonic stem cells (ESCs), progenitor cells, or induced pluripotent stem cells (iPSCs) to tissue or organ explants [6,7,8]. Human pluripotent stem cells (hPSCs), including human embryonic stem cells (hESCs) and human-induced pluripotent stem cells (hiPSCs), are cells with the ability to self-renew and develop into all cell types in a human adult body [9]. hPSCs can produce valid in vitro models for dissecting disease mechanisms, discovering novel drug targets, screening drug candidates, and evaluating drug safety and efficacy [10]. ESCs facilitate research on mammalian neuronal development, neurodegenerative disorders, and regenerative therapies [11]. Three-dimensional technology can be used to model human organ development and several human diseases “in a dish” [12]. In addition, evidence indicates that patient-derived organoids can predict drug responses to cancer [13]. Indeed, biobanks of patient-derived tumor organoids are used in drug development research, and they are promising for evolving personalized and regenerative medicine [14,15,16,17].

The road map for our exploring the brain and the retinal organoids relies on the fact that brain and retinal tissues, due to neuronal activity, are the main energy-demanding systems in the body [18]. Other principles on which our reasoning is based regarding the common embryological origin [19,20,21] and the now understood intertwining between brain disorders and eye diseases [22]. Studies of Alzheimer’s disease (AD) and glaucoma have demonstrated neurodegenerative changes and disease traits in both brain and eye regions [22,23,24,25]. Remarkably, anatomical and functional alterations, such as the thinning of the ganglion cell and retinal nerve fiber layers [26], the presence of protein aggregates, and glial activation [27,28,29] can be detected in the postmortem evaluation of the retina in AD patients and animal models, thus strengthening the idea that the retina could be exploited in early AD diagnosis. Moreover, visual deficits, including difficulty reading [30], depth perception [31], and color recognition [32], are also reported in the early stages of AD. Changes in retinal structure and function have been reported in other neurodegenerative disorders, such as Parkinson’s disease (PD) [33].

We review here how the brain and retinal organoids have been developed and analyzed technically. They have been used as tools for modeling neurological and eye diseases, as well as considering their potential for translational applications. Furthermore, this review is focused on the importance of blood–brain barrier (BBB) and blood–retinal barrier (BRB) in vitro models as reliable and predictive screening tools for BBB, and BRB-penetrating compounds, also discussing shortcomings, limitations, and advantages of developed 3D cultures.

## 2. Brain Organoids to Investigate Brain Disorders

Protocols to generate brain organoids have already been reviewed and modified by several studies, often with overlapping approaches, highlighting factors that are most relevant for an effective differentiation [34,35]. The serum-free culture of embryoid bodies (SFEBq) method to generate 2D and 3D neuronal cell types from ESCs has been widely used [36,37,38]. In several brain organoid protocols, basement membrane matrices, such as Matrigel, have been applied as it has been shown to be an effective scaffold [37]. However, these matrices show some problems: batch-to-batch variability during manufacture, complicated imaging, risk of contamination, and high cost [34,35].

Recently, Gabriel et al. [39] have demonstrated that hiPSC-derived brain organoids assemble forebrain-associated bilateral optic vesicles (OV), which form progressively as visible structures. These OV-containing brain organoids (OVB-organoids) constitute a developing OV’s cellular components, including primitive corneal epithelial and lens-like cells, retinal pigment epithelia, retinal progenitor cells, axon-like projections, and electrically active neuronal networks. OVB-organoids developed both neural and non-neural cell types. Interestingly, these OVB-organoids are light-sensitive. Thus, brain organoids have the intrinsic aptitude to self-organize forebrain-associated primitive sensory structures in a topographically restricted way and can allow interorgan contact studies within a single organoid. In future studies, strategies could be applied to keep the OVB-organoids viable to examine mechanisms that cause retinal disorders [39].

Region-specific neural organoids can be created by the differentiation of hiPSC [40]. These organoids can recapitulate distinct brain regions that arise during human brain development, as it has been shown in cortical plate [41], forebrain [42], midbrain [43], and hypothalamic organoids [44]. These self-assembly platforms can mimic some features of human brain development, for instance, topological organization analogous to human tissue and can even create functionally mature brain cells synaptically connected [42]. Although human brain organoids can be used to answer many questions, there are some limitations, related to ESCs-derived organoids, since they more closely resemble immature brains and are not suitable to model specifically old age diseases, such as AD; nevertheless, it is an approach to investigate an array of otherwise difficult to study conditions such as neurodevelopmental handicaps, genetic disorders, and neurologic diseases [45]. Then, a more specific protocol should be applied to study specific brain regions [46,47].

Brain organoids represent a powerful in vitro approach to model neurodevelopmental, psychiatric, and neurodegenerative diseases (Table 1).

Many researchers developed cerebral organoid models that generate functional cortical neurons and can sum up forebrain, midbrain, and hindbrain areas with functional electrophysiological properties to probe the neurodevelopmental mechanisms of autism spectrum disorder (ASD), excessive growth of the fetal brain or macrocephaly [48], and microencephaly (MCPH) [37,49,50]. Severe MCPH in infants born has been correlated to prior/premature exposure to Zika virus (ZIKV). Brain organoids are a powerful tool to analyze rapidly the effects of ZIKV on human brain development, providing insight in a short time [49]. Several studies reported an effect of ZIKV on brain organoids [46,51,52,53,54,55]. ZIKV infection leads to increased cell death and reduced proliferation, resulting in decreased neuronal cell-layer volume resembling MCPH [46]. In addition, Garcez et al. [53] found that ZIKV reduces cell viability and growth in neurospheres and brain organoids, suggesting that ZIKV abrogates neurogenesis during human brain development [49]. Moreover, iPSCs-derived cerebral organoids from a patient with abnormal spindle-like primary microcephaly (ASPM) could recapitulate neurogenesis abnormalities in the disease [50].

In addition, for studying the mechanism underlying diseases involving well-defined brain malformations, organoids derived from hPSCs bearing causal mutations for neurodevelopmental and psychiatric diseases [56,57,58,59,60,61] can be used to identify previously unknown abnormalities. Brain organoid models sum up aspects of neurodegenerative diseases, including AD [62,63,64] and PD [65,66,67,68], and exploring the utility of these models for therapeutic applications is critical. The generation of models based on disease-specific iPSCs simplifies the progress toward studying Huntington’s disease (HD) and screening potential treatments. However, the neurodegenerative process of HD does not affect just a single population of cells but different tissue types. The resolution of this issue could be the generation of brain organoids [69], which has already been successfully performed by Conforti et al. [70].

Concerning translational applications, the use of brain organoids has been explored only in hPSCs models of ZIKV and congenital brain malformation (CBM) [37,46,55], and not in mice [37] or primate [71] PSCs models. This is because the translational applications of brain organoids are limited by high heterogeneity between cell lines, prolonged culture times, and laborious procedures. Studies attempting to standardize organoid differentiation have found that cell line-specific variables influence differentiation efficiency during the initial stages [72].

## 3. Retinal Organoids for Disease Modeling Application

Taking advantage of the original retina organoid protocol [73], retinal organoids that closely resemble the retina using mouse and human stem cells following several differentiation procedures were developed [74,75,76,77,78]. These organoids largely recapitulate major cellular and molecular events of in vivo retinal morphology and retinogenesis, with appropriate apical-basal polarity and time-dependent self-patterning of major cell types into a laminated structure [73]. However, incomplete functional maturation of photoreceptors in culture is still present after differentiation protocols as consequence retinal organoids are generated with no light responses and undeveloped outer segment-like structures [74,76,77,78,79,80]. Recent differentiation protocols are time consuming and labor intensive for isolating OV-like structures from adherent cultures by dissection [81]. This impedes applications that require large-scale production of retinal organoids, for example, biochemical studies and high-throughput drug screening. Overcoming this issue, Regent et al. [82] have reported a simple and efficient technique for generating retinal organoids by scraping the entire adherent cell culture and growing the resulting cell aggregates in a free-floating condition. Following this procedure, retinal organoids were often generated with the retinal pigment epithelium (RPE), and OV developed morphologically well-defined and was harvested easily in a few days. A successful protocol was also designed by Cowan et al. [83], who developed light-sensitive human retinal organoids with three nuclear and two synaptic layers, functional synapses from iPSCs, and their production was in large quantities. Single-cell transcriptomes from cells dissociated from developing human multilayered organoids revealed progressive maturation of retinal cell classes and showed that matured organoids reached a stable “developed” state in vitro at a rate similar to human retina development in vivo [83].

Since the retina can be defined as the window to the mind, the dysfunction of retinal neurons in age-related macular degeneration (AMD), glaucoma, or diabetic retinopathy (DR) is a major cause of blindness. Hereditary eye degenerative diseases, such as retinitis pigmentosa (RP) and Leber congenital amaurosis (LCA), are genetically and clinically heterogeneous conditions that lead to progressive loss of vision and blindness as the outcome, which is one of the most feared disabilities, as effective treatments do not currently exist [84]. Retinal organoids derived from hPSCs have significantly improved our tools to study human development and eye degeneration diseases such as RP, AMD, and LCA in the dish (Table 1), as reported by many studies [5,77,85,86,87,88,89,90]. Several studies suggest retinal organoids with eye cup-like structures may provide insight into developmental and regenerative processes [88]. Moreover, retinal organoids carrying eye disease-causing mutations could potentially recapitulate disease progression in vitro and facilitate the development of effective treatments [91].

Currently, insufficient therapeutic possibilities are available due to the physiologic differences between human and animal models, and the lack of efficient in vitro systems. Regardless, retinal organoids have been used for cell replacement therapy studies [92,93,94]. Stem cell transplantation studies showed that this therapy is a promising approach to restore visual function in eyes with degenerative eye diseases such as RP, AMD, and Stargardt’s macular dystrophy. For translational applications, orthotopic transplantation of retinal organoids has been explored both in murine [95] and in primate [96] models of eye degeneration. However, the translational applications of retinal organoids are limited by high heterogeneity between cell lines, prolonged culture times, and laborious procedures. Studies attempting to standardize organoid differentiation have found that cell line-specific variables influence differentiation efficiency during the initial stages [81]. Furthermore, approaches to handling organoids over these prolonged culture periods determine the efficiency of maturation at later stages [97].

Given these issues, it is of the utmost importance to ensure that the initial differentiation and generation of retinal organoids are properly accomplished. The maintenance and maturation of organoids are consistent and reproducible [98]. In accordance, other reviewers highlighted the limitations of retinal organoid technologies [99,100,101]. Despite initial successful attempts at modeling inherited retinal dystrophies [102], the high complexity and low yield in current protocols remain substantial technical challenges [100], and many questions remain still open [11,99].

## 4. Brain and Retinal Organoid Research: Analytical Techniques

Organoid studies have mainly relied on phenotypic readouts thus far (that is, aspect, shape, and number of organoids) [72]. In Table 2, we summarize the analytical techniques applied up to now to brain and retinal organoids research: image-based analysis [37,43,46,48,65,86,103,104,105]; protein determination and quantification [106]; gene analysis [103,107,108,109]. 

These techniques range from optical observation, which is the most powerful and oldest technique in biological research, to gene expression, less explored in the brain and retinal organoid development. Collectively, the data showed different limitations, reduced throughput, and increased cost for certain techniques that should be improved (e.g., gene analysis).

## 5. Comparison of Blood–Brain and Blood–Retina Barriers: Structure, Homeostasis, Damages and Permeability

The BBB is a dynamic interface that plays a key role in the homeostasis maintaining of the central nervous system (CNS) [110]. BBB defines the exceptional properties of the microvasculature of the CNS. The cerebral blood vessel formed by ECs is an essential element of the BBB [111]. The ECs layer has continuous intercellular tight junctions (TJs), and it is not fenestrated. Therefore, the movements of molecules, ions, and cells are extremely low through them and limited by a series of specific transporters, which allow delivery of nutrients to the brain and extrusion of potential toxins and pathogens [111,112]. The development and conservation of the BBB are governed by interactions with different vascular, immune, and neural cells [113]. Astrocytes, pericytes, and extracellular matrix (ECM) elements offer both structural and functional support to the BBB. In addition, a dynamic functional unit is represented by the neurovascular unit (NVU), which refers to neurons, microglia, and peripheral immune cells that likewise participate in this cellular interplay [114]. Even if this heavily restricting barrier capacity allows BBB ECs to regulate CNS homeostasis, it provides an obstacle for drug delivery to the CNS, and thus, major efforts have been made to create methods to modulate or bypass the BBB for delivering therapeutics [115]. Experimental and clinical evidence show that BBB dysfunctions can induce ion dysregulation, altered signaling homeostasis, as well as access of immune cells and molecules into the CNS. These processes lead to neuronal dysfunction and degeneration with increased susceptibility for AD, PD, HD, and amyotrophic lateral sclerosis (ALS), brain tumors, epilepsy, stroke, and glaucoma [110,111,112,113,116,117,118]. Like the BBB, the BRB also plays an essential role in maintaining the health of the CNS. These two systems have shown partially overlapping roles in the postnatal brain and retinal vasculatures [119]. The BRB is indispensable in maintaining an appropriate environment for optimal retinal function [120]. Indeed, vasculature and BRB alterations are extensively reported in the AD retina, and their investigation as possible diagnostic tools is under evaluation [121,122,123]. Retinal tau protein plays a key role in regulating axonal transport and signaling in the retina [124]. Reduced clearance of retinal beta-amyloid (Aβ) and other neurotoxic substances contribute to BRB dysfunction and breakdown [125], inducing a persistent inflammatory state [126,127].

BRB is composed of both an inner (iBRB) and an outer barrier (oBRB), whose key differences were previously summarized by [128]. While the iBRB is formed by TJs between retinal capillary endothelial cells (RCECs), the oBRB is composed of TJs and RPE, which separate the neural retina (NR) from the fenestrated vascular system of the adjacent choroid plexus. This oBRB regulates the molecular movements of solutes and nutrients from the choroid to the sub-retinal space. In contrast to the oBRB, the BBB is established by ECs rather than by epithelial cells. In the brain, ECs differentiate in a CNS-specific manner under the stimulus of astrocytes [129], whereas RPE cells in the eye are able to produce barrier features in the absence of astrocytes [130]. However, in both cases, the expression of blood-barrier markers such as HT7-neurothelin and the endothelial barrier antigen (EBA) is extremely upregulated when barrier function is established during development [131]. On the contrary, the iBRB, like the BBB, is localized in the inner retinal microvasculature and includes the microvascular endothelium, which lines these vessels. The TJs situated between these cells induce extremely selective diffusion of molecules from the blood to the retina, and the barrier is crucial to preserve retinal homeostasis. The retina has the highest oxygen consumption per weight of any tissue in the body and the BRB (both outer and inner) is essential in providing certain nutrients to maintain this high metabolic rate [128].

TJs in both the iBRB and the oBRB are complex dynamic structures. In the context of these barriers, the integrity of these TJs is decisive to sight [128]. Indeed, an accumulation of blood-borne proteins and other possibly toxic solutes within the retina can be induced by damage to either of these barriers [132]. In particular, the disruption of the oBRB increases the incidence of ocular pathologies, such as DR and diabetic macular edema (DMO), AMD, central serous chorioretinopathy (CSCR), Sorsby’s fundus dystrophy, and RP [128,132].

Even if epithelial oBRB and endothelial BBB have developed as separate entities with many site-specific functions, their transport and permeation features show surprising similarities that consist of the polarized expression of the two major efflux pumps belonging to the ATP-binding cassette (ABC) family of transporters: multidrug resistance protein (P-gp) and multidrug resistance-associated protein (MRP) [131]. Moreover, differences were reported in ABC-transporter expression/function at the BBB and the BRB. The pharmacokinetics and pharmacodynamics of drugs targeting the brain and retina may differ in this regard [133]. As with the BBB, lipophilic substances showed high permeabilities also across the BRB through passive diffusion [134,135]. Lipid-soluble (lipophilic) compounds with low molecular weight (MW) and positive charge can cross the BBB [136]. MW is an important parameter in determining the free diffusion of small molecules across the BBB as well as the BRB. Once the MW is >400 Da, the BBB permeability of the drug does not increase in proportion to lipid solubility; indeed, the largest (500 kDa) molecules fail to penetrate the brain. An increase in the surface area of a drug from 52 Å^2^ (e.g., a drug with an MW of 200 Da) to 105 Å^2^ (e.g., a drug with an MW of 450 Da) dramatically decreases its BBB permeation [137]. In addition, the compounds with the ability to cross the BBB should have a log[brain]/[blood] (logBB) ≥ 0.00 [138]. Recently, a dataset for modeling BBB permeability of small molecules providing some physiochemical similarities and differences was published [139]. Several biomarkers that can help to assess the BBB permeability and integrity in vitro or in vivo are reported [140]. Unfortunately, a very limited number of studies have been conducted to obtain experimental data on BRB permeability. Also here, lipophilicity, extremely small MW, and charge are the main physical-chemical parameters that determine the highest RPE permeability [141,142,143]. There are few studies providing the permeability coefficients of the RPE [131,142,144]. For instance, the smaller MW and lipophilic drug lidocaine (288.8 Da, log P = 1.54) revealed the highest permeability, whereas the larger molecular weight and hydrophilic drug ciprofloxacin (367.8 Da, log P = −0.54) exhibited the lowest permeability [145]. On the contrary, larger lipophilic molecules and hydrophilic molecules require ATP-dependent transports to cross the barrier, including receptor-mediated vesicular transport, non-receptor-mediated pinocytosis, transporters, and pumps [18,135]. Pharmacokinetic aspects of retinal drug delivery were reviewed detailly by Del Amo and collaborators [146].

## 6. Advances in BBB In Vitro Modeling: Organoids

In vitro BBB models are crucial tools for optimizing the transport of drugs across the BBB. They are also crucial for developing new drugs that reach the brain, and for predicting which compounds would be effective in treating neurological diseases. In vitro BBB modeling has been in development since the 1980s [147]. However, reproducing key BBB properties ex vivo remains challenging.

Many researchers have widely used the static 2D Transwell because it is the simplest system to represent the BBB. In the most commonly used Transwell system, the ECs are usually grown in the upper (luminal) compartment of the Transwell in a cell-specific growth medium. Additional cells, such as astrocytes or pericytes, are normally cultured on the lower (abluminal) side of the membrane brain [148,149]. Even if this mid- to high-throughput model offers versatility and ease of culture [148], it has been criticized because of difficulties of preserving reproducible BBB function and properties [149]. Despite several well-known limitations, the brain ECs grown in culture are still used to model the BBB.

As stem cell-derived brain ECs are difficult to obtain, immortalized human cell lines such as human cerebral microvascular endothelial cells (HCMEC)/D3 are often preferred for a human model BBB [150]. In addition, the cell line primary human brain microvascular endothelial cells (HBMECs) has been shown to give the best barrier properties for permeability studies using Transwells [151]. First, human cells should be used to diminish species-specific answers, though they are not often used, as immortalized human cell lines do not produce an adequately tight barrier [152,153]. Despite this limitation, HCMEC/D3 and HBMECs continue to be used to identify changes in barrier integrity by measuring relative values before/after treatment or disruption [151,154]. Immortalized cell lines are an attractive option due to their low cost, ease of use, and their ability to be passaged multiple times while retaining BBB transporter expression [150]. Urich et al. [155] reported the successful assembly of human primary astrocytes, pericytes, and ECs into a BBB spheroidal model. A similar model was investigated by Cho and collaborators [156], co-culturing primary human astrocytes and human brain vascular pericytes (HBVPs) with two different human brain EC types: primary HBMECs and immortalized HCMEC/D3. In accordance, Bergmann et al. [157] described a triple co-culture of HBVPs, primary human astrocytes, HCMEC/D3 cells, or primary HBMECs under low-adhesion conditions into a multicellular structure to obtain BBB organoids. These organoids can accurately mimic the BBB since they display enhanced BBB features (e.g., molecular transporters, expression of TJs, and drug efflux pumps) as compared with those of ECs cultured in the Transwell system [156]. Concentrating on the drawbacks of the conventional organoid in vitro BBB model, the development of a 3D spheroid of BBB has been successfully reported by Nzou and coworkers [158], proposing a model that closely mimics the human brain tissue since it is comprised of six cell types found within the brain cortex. These cell types include HBMECs, HBVPs, human astrocytes (HA), human microglia (HM), human oligodendrocytes (HO), and human neurons (HN), with ECs enclosing the brain parenchymal cells. In addition, Nzou et al. [158] validated the expression of TJs, and transport proteins showed that this model can be used in toxicity assessment studies for molecules that have the potential to cross or open the BBB. Despite the current advances in the development of BBB spheroids and organoids, they usually lack essential elements of the BBB cellular milieu, including microglia, six distinct cortical layers, and endothelial vasculature. Moreover, the limited formation of microglia and mature neurons limits its utility for specific in vitro neurological disorders models [159].

Microfluidic devices have been developed to further improve the physiological characteristics of the BBB in culture. The efforts to produce a more dynamic and realistic representation of the BBB morphology in a living system by reproducing the microcirculatory environment in the brain to account for blood flow and shear stress have induced the development of the hollow fiber dynamic in vitro BBB model [160] and microfluidic BBB systems [161,162]. However, these devices are also incomplete in terms of throughput, and their construction is rather complex, making them moderately unreachable to many laboratories. One of the recent and most promising approaches is the development of hiPSC-derived neuronal cultures that can “self-assemble” within microfluidic devices. Therefore, they promote neurite outgrowth and interaction with other neural cell types and enhance synaptic connections [163]. These so-called “organs-on-a-chip” (OACC) are set to revolutionize drug discovery [164]. Park et al. [165] developed a microfluidic BBB-chip model from hiPSCs that maintains relevant human physiological features for a week, presents permeability restriction that lasts up to 2 weeks, has high levels of expression of TJs proteins, and appropriate function of efflux proteins. The group confirmed that the BBB chip was able induce transporter-mediated drug efflux, including suitable substrate specificity, and they tested CNS-targeting peptides, nanoparticles, and antibodies crossing the BBB, demonstrating the BBB chip could test clinically relevant compounds [165].

Finally, in vitro BBB models may be critical to the screening and development of novel and effective therapeutics against many neurological disorders, and a valid one to three cell type models have been described [166,167]. Recently, a summary of how these in vitro models of the BBB can be applied to the study of human brain diseases and their treatments was extensively reported by Williams-Medina et al. [168]. The latter have chosen NeuroAIDS, COVID-19, multiple sclerosis, and AD as examples of in vitro model application to neurological disorders. For modeling neurodegeneration in vitro, the following methods could be applied in NVU/BBB models: (i) exposure of cultured cells to Aβ in vitro to reproduce amyloid-mediated acute cytotoxicity; (ii) isolation of cells from the brain of transgenic mice with AD genotype for further co-culture and examination; (iii) isolation of cells from the brain of animals with non-genetic in vivo models of AD (i.e., intrahippocampal injection of Aβ); (iv) establishment of mixed models consisting of organotypic culture obtained from the animals with AD model and cells (i.e., BMECs) from the intact animals; (v) application of genome editing or reprogramming technologies to get the in vitro model with the desired morphological and functional modifications resembling those in Alzheimer’s type neurodegeneration [169].

The inconsistent results across animal models of neurological diseases and their impact on human studies [170,171] suggest that BBB organoids could provide an effective alternative.

## 7. Conventional In Vitro Models and Organ-on-a-Chip for Innovative BRB

Conventional in vitro models of BRB are important tools allowing us to clarify the mechanisms involved in retinal pathophysiology as well as the tracking occurring in the barrier [172,173]. The BRB is an interface extremely controlled that separates the circulation from the retinal tissue [120]. To resemble this interface Transwell, as in vitro models of BRB inserts were applied. The Transwell inserts are permeable supports on which cells are seeded, and that include an apical and basal chamber [174]. Additionally, these planar models allow us to quantify easily barrier properties through permeability to fluorescent tracers and transepithelial-transendothelial electrical resistance (TEER) [175]. In the standard use of Transwell inserts to model the BRB, retinal ECs for the iBRB or RPE for the oBRB were seeded on the upper compartment of the Transwell to create monoculture devices or integrated as tri-culture devices where the other cell types are sown on the opposite side of the insert and-or at the bottom of the well [176,177,178]. Indeed, co-cultures of BRB are extensively used to understand the cross-talk between the cells of the retinal unit. Based on the importance of communication between cells, studies have reported that the integration of certain cells could influence ECs activities and BRB permeability, also allowing to explore developmental, functional, and pathological processes of the retina. Recently, a BRB in vitro model closer to the human in vivo environment was obtained by co-culturing human retinal endothelial cells, human retinal pericytes, and human retinal astrocytes [179]. For instance, integrating pericytes, astrocytes, and-or astrocyte-conditioned medium with ECs in an iBRB Transwell model enhanced TJs proteins and TEER values compared with monoculture and provided a more relevant frame to investigate permeability [180,181]. While in an oBRB model, the coculture of ECs-RPE reduced the RPE barrier properties, this disruption of barrier occurs in ocular pathologies, such as choroidal neovascularization [178].

In vitro models of BRB have mostly been established using primary cells isolated from animal or from human samples to increase model relevance to clinical diseases [176,177,178]. Human immortalized cell lines, such as the RPE cell line ARPE-19, have been developed to improve availability and robustness [182]. In addition, iPSCs have been developed as a source to produce retinal cells because of their self-renewal capacity, potential to differentiate into different lineages, and to create vascular progenitors and ECs [74,183,184,185].

As reviewed by Ragelle et al. [174], conventional in vitro models of BRB can be improved through OACC systems. They consist of micrometer-sized devices that allow the culture of cells under perfusion and, in a spatially precise microenvironment, mimic tissue or organ physiology. A suitable BRB-on-a-chip should validate appropriate barrier properties with the formation of TJs, reproducible permeability to reference compounds, and medium-to-high throughput screening capacity [174]. These devices completely micro-engineered have several advantages: flexibility of design features, the possibility of integrating analytics directly within the chips, and high-resolution imaging. Finally, the microscale reduces the use of reagents and cells, permits a media-to-cell ratio closer to physiological values, allows analytical sampling in small volumes, and favors high-throughput experimentation [186]. Thus, BRB-on-a-chip represents a powerful in vitro platform in ophthalmic drug discovery and development [174]. Except for OV-containing brain organoids (OVB-organoids) reported by Gabriel et al. [39], no data have been published about BRB organoids.

## 8. Perspectives: Biobanks

Organoids can be stored in biobanks and used for basic research, organ transplantation, drug formulation testing [14,16], as well as regenerative medicine [15,16]. Generating organoid biobanks is crucial for personalized medicine as it brings the ability to perform high-throughput drug screening, epigenomic and transcriptomic analysis, and copy number variations of individual patients at a large scale [187]. While biobanks have been generated for different tumor-derived organoids [188], these can be extended to the development of organoid biobanks with individual disease variants derived in ESCs, or from iPSCs of rare diseases. Since brain organoids were recently identified as a promising living biobank resource for neuroscience research [189], we hope to see also a retinal organoid biobank in the future that will accelerate personalized drug development in the ophthalmology field. From the future perspective of both these organoids, we propose the potential applications of organoid culture tools for the advancement of biological research (Figure 1).

Patient-derived brain and retinal 3D organoids have provided new insights into disease modeling and have opened new possibilities for personalized medicine [190].

## 9. Discussion

Organoid systems leverage the amazing self-organizing properties of stem cells to re-create complex tissue and organ development in a dish. In vitro organoids are extremely attractive for broad applicability, ranging from understanding the basic developmental dynamics to drug treatment personalization or autologous cell therapy. This is because of their proximity of cell-type composition, structural organization, and functionality to the respective in vivo tissues [1,12,15]. An obvious advantage of organoid cultures for disease modeling, compared with traditional cell cultures of a single cell type, is their ability to mimic large quantities of pathologies by recapitulating specific human features that could be relevant for translational studies [191]. Brain organoids represent a powerful in vitro approach to model brain development [35,108,192], understand neurodevelopmental diseases [193], recapitulate aspects of neurodegenerative diseases [194,195,196,197], and for personalized drug screening when an individual’s hiPSCs are used [49,198]; while retinal organoids have been reported as human eye disease models, pharmaceutical testbeds, and cell sources for transplantations [5,82,84,98,100,199,200].

However, the limitations of current organoid systems are several: high culture costs; limited level of maturity and function; the limited lifespan of organoids is often a direct consequence of restricted accessibility; readouts or measurements are technically challenging in 3D organoids compared to the standard technique in 2D Transwell culture systems to assess barrier integrity; heterogeneity in organoid formation efficiency, end-point morphology, and function; variabilities in the organoid generation, which often require multiple experimental steps [72]. Given the lack of reproducibility, novel stem cell-based differentiation approaches are necessary [72]. One of the primary factors limiting further development of organoid technology has been size restriction imposed by insufficient nutrient delivery to the organoid interior due to the absence of vascularization [201,202], which is especially true for brain organoids [193]. A lack of vascularization additionally prevents the modeling of critical aspects of brain physiology, such as the BBB [203]. Similar to the BRB, the BBB can act as a checkpoint to the transit of many drugs, and for these reasons, in vitro vascularization of the brain and retinal organoids using ECs might contribute to fostering the identification and development of new molecular targets [22]. It is well recognized that in vitro models of BBB and BRB could be used as tools in translational medicine [168,174]. However, no data have been reported yet, because of many limitations of traditional brain organoid transplant. This is also true for other organ transplant procedures with high demands and low success rates, such as renal transplants [2]. Recently, a few studies showed the potential formation of OVB organoids, but not BRB organoids. Based on the structure and permeability similarities between the iBRB and BBB and the recent technical advantages, we are confident that as for BBB, also BRB organoids could be created in the future to recapitulate the key BRB properties and functions. Thus, the development of BBB and BRB in vitro studies became extremely important as these barriers play a role in both brain and retinal health and disease. Additionally, BRB- and BBB-on-a-chip have been developed as microfluidic cell culture devices to overcome the limitations of static in vitro models. Modifying the architecture of the device allows the recreation of the physiological environment in vivo while measuring barrier function. The assessment of barriers in organs-on-chips can be difficult, but they offer the opportunity of continuous, non-invasive sensing of barrier quality, which allows better investigation of central aspects of pathophysiology, biological processes, and progress of therapies that target barrier tissues [204]. The development of more accurate and sophisticated barriers-on-a-chip with the capacity to grow in vitro connected with appropriate vascular supplies and nerves, paves the way for the development of functional and integral in vitro BRB and BBB models and offer a promising avenue by enabling future research scientists to perform experiments on a realistic replica when testing the effectiveness of novel experimental therapies [168,174].

In this review, we summarized the analytical methods applied in the brain as well as in retinal organoids research and, based on the disadvantages (e.g., reduced throughput and difficult sample preparation), we assess the necessity of improving certain techniques, which, in turn, allow accurate disease modeling. We review that these organoids are effective in vitro tools for disease modeling: while brain organoid technology has greatly enhanced neurodegenerative and neurodevelopmental disease and psychiatric disorders research [193,205], also the retinal organoid has improved visual research [5,200]. Indeed, we highlight that organoids from hPSCs-based retinal and brain organoids provide an outstanding opportunity to explore cellular and subcellular functions within in vitro models that closely recapitulate the native 3D configuration of the human neural tissue [22]. Theoretically, brain and retinal organoids offer a potential alternative to cell and whole organ transplantation by providing autologous tissue. However, the lack of studies in which brain and retinal translational applications have been carried out does not allow to make a definitive conclusion about their roles and potentialities in neurological/retinal disease research [37,46,55].

## 10. Conclusions

To conclude, there are still many obstacles to overcome before iPSC-derived technology can be used directly in retinal degeneration and neuronal diseases, as well as in translational studies. These in vitro platforms offer promising tools to develop novel in vitro therapeutic approaches [165,206]. Large cohort iPSC-based studies could be allowed by biobanking, which can significantly drive iPSC-based therapeutic applications in the future [207].

## Figures and Tables

**Figure 1 cells-11-01120-f001:**
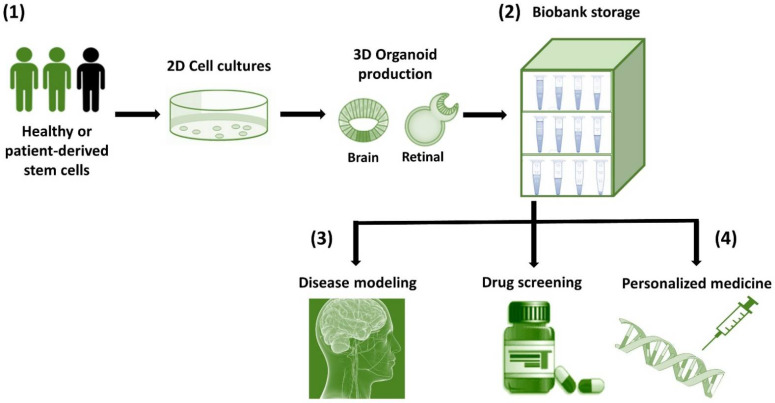
Schematic representation of the workflow from the production of brain and retinal organoids to their possible conservation/biobanking and potential applications. (1) Healthy or patient-derived stem cells differentiate and produce brain and retinal 3D organoids. (2) Biobanking, whereby samples obtained from patients can be used to store patient-generated organoids as a resource for future research. (3) Disease modeling, to understand the mechanisms of human diseases such as neuronal disorders and retinal degeneration through various laboratory techniques. (4) Drug screening and personalized medicine, in which patient-derived organoids can be used to predict drugs response and as resources for regenerative medicine coupled with genetic engineering.

**Table 1 cells-11-01120-t001:** Human pluripotent stem cells-derived brain and retinal organoids for modeling diseases.

Tissue/Organ	Source	Disease Modeled	References
Brain	hPSCs	Microcephaly primary hereditary	[37,49,50]
		Zika virus, congenital brain malformation	[46,49,51,52,53,54,55]
		Autism spectrum disorders/macrocephaly	[48]
		Rett syndrome	[56]
		Miller-Dieker syndrome	[57,58]
		Sandhoff disease	[59]
		Schizophrenia	[60,61]
		Alzheimer’s disease	[62,63,64]
		Parkinson’s disease	[65,66,67,68]
		Huntington’s disease	[69,70]
Retinal	hPSCs	Retinitis pigmentosa, age-related macular degeneration	[5,77,85,86,87,88]
		Leber congenital amaurosis	[88,89]
		Glaucoma	[90]

Abbreviations: hPSCs, human pluripotent stem cells.

**Table 2 cells-11-01120-t002:** Analytical techniques in brain and retinal organoids research.

Analytical Techniques	Physical and Technical Limitations	Advantages	Disadvantages	References
**Image-Based Analysis**				
Histochemistry Histology and immunostaining Immunofluorescence	Destroying technique; rigorous requirement for fixing and cutting of tissues	Consolidated procedure; simple imaging	Reduced throughput and automatization; a restricted set of standard stains	BOs: [37,103]
Electrophysiology	Only for electrically active cells such as neurons or (photo)receptors	Functional valuation; cells intrinsic properties data	Reduced throughput and difficult sample preparation	BOs: [43,46,48,65] ROs: [86]
Light-sheet imaging	Concomitant imaging of several organoids not allowed because the small sample size	Appropriate for live imaging; 3D data	Reduced throughput and difficult sample preparation; restricted to one condition	BOs: [104] ROs: [105]
**Protein Determination And Quantification**				
Immunoassays (ELISA, WB)	Destroying technique	Functional data (proteins amount and interactions, PTMs); high sensitivity (ELISA) as well as specificity (WB)	Reduced automatiza-tion; labor-intensive; no 3D data	BOs: [106]
**Gene Analysis**				
qRT-PCR	Destructive method; mRNA levels are only a proxy for the functional state of a cell	Quantitative gene expression levels, high-sensitivity	No data of protein quan-tities	BOs: [103]
Gene expression and RNA sequencing	Destroying technique; scRNAseq necessitates pure single-cell preparation	Entire transcriptome data; scRNAseq has single-cell-level resolution	Expensive; reduced throughput; expertise required about study and processing	BOs: [107,108,109]

Abbreviations: BOs, brain organoids; ELISA, enzyme-linked immunosorbent assay; PTMs, post-translational modifications; ROs, retinal organoids; scRNAseq, single-cell RNA sequencing; WB, Western blot.

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
