# Peer review of "Brain and Retinal Organoids for Disease Modeling: The Importance of In Vitro Blood–Brain and Retinal Barriers Studies"

_cells, 2022, doi:10.3390/cells11071120_

Round 1
Reviewer 1 Report
The manuscript is nicely written in fluent English. The information contained in the manuscript are of relevance for the people that are performing research in the field.
In my opinion the manuscript is suitable for publication in the present form in Cells.
Minor suggestions to be implemented are reported below.
In my opinion this review article could be informative not only for the pharmacologist and the biologist who develop different models of diseases, but in general for all those involved in drug discovery, in particular for the medicinal chemist who develop and synthesise new molecular entities. In this frame, as the Authors frequently compared brain vs retinal models for the barriers, it would be interesting to report, if any, the differences in physico-chemical requirements for compounds to cross retinal and brain barriers and comparing them. Maybe a table listing and comparing hydrophobicity, polar surface area, molecular weight values (among others) that an ideal compound should have for crossing brain or retinal barriers would add further key information.
One minor inconsistency was noted. In the introduction and in the conclusion, the Authors state that the model derived from embryonic stem cells are probably still not well suited for aging diseases, as they mainly reflect the features if immature brain. This assumption may seem dissonant with the sentence at line 120, please rephrase.
The phrase at lines 225-226 is misleading, please rephrase.
Author Response
Reviewer 1 Reply
-In my opinion this review article could be informative not only for the pharmacologist and the biologist who develop different models of diseases, but in general for all those involved in drug discovery, in particular for the medicinal chemist who develop and synthesise new molecular entities. In this frame, as the Authors frequently compared brain vs retinal models for the barriers, it would be interesting to report, if any, the differences in physico-chemical requirements for compounds to cross retinal and brain barriers and comparing them. Maybe a table listing and comparing hydrophobicity, polar surface area, molecular weight values (among others) that an ideal compound should have for crossing brain or retinal barriers would add further key information.
We thank the reviewer for the observation. We changed the structure of paragraph 5. “Comparison of Blood-Brain and Blood-Retina Barriers: Structure, Homeostasis, Permeability and Damages”, focusing at the end on the comparison between physicochemical requirements for compounds able to cross brain and retinal barriers (lines 283-301). We decided to amplify this section and support through some examples the mutual physicochemical characteristics among the small molecules (drugs) with the highest BRB and BBB permeability. We amplified instead of creating a new table, as recently Meng et al. (A curated diverse molecular database of blood-brain barrier permeability with chemical descriptors. Sci Data. 2021 Oct 29;8(1):289. DOI: 10.1038/s41597-021-01069-5) present a new Blood-Brain Barrier database, listening several small molecules compounds and their physicochemical properties for modeling BBB permeability in terms of similarities and differences between BBB+ and BBB− compounds. A new table would have been redundant and unbalanced since, on the contrary, a limited number of studies have been conducted to obtain experimental data of BRB permeability. Thus, we added new references to amplify this aspect.
-One minor inconsistency was noted. In the introduction and in the conclusion, the Authors state that the model derived from embryonic stem cells are probably still not well suited for aging diseases, as they mainly reflect the features if immature brain. This assumption may seem dissonant with the sentence at line 120, please rephrase.
We thank the reviewer for the observation. We moved the sentence, and we rephrased it as follows: “Although human brain organoids can be used to answer any questions, there are some limitations in particular related to ESCs-derived organoids, since more closely look like immature brains, they cannot be suitable to model old age diseases, such as AD, but it is nevertheless an approach to investigate an array of otherwise difficult to study conditions such as neurodevelopmental handicaps, genetic disorders, and neurologic diseases (add ref. 45. Boisvert EM, Means RE, Michaud M, Thomson JJ, Madri JA, Katz SG. A Static Self-Directed Method for Generating Brain Organoids from Human Embryonic Stem Cells. J Vis Exp. 2020 Mar 4;(157):10.3791/60379. DOI: 10.3791/60379)” (Line 93-97).
-The phrase at lines 225-226 is misleading, please rephrase.
We thank the reviewer for the correction. We moved the sentences at the end of paragraph 5. [Comparison of Blood-Brain and Blood–Retina Barriers: Structure, Homeostasis, Permeability, and Damages], and we rephrased also based on the reviewer’s first request, as follows: “Lipid-soluble (lipophilic) compounds with low molecular weight (MW) and of positive charge can cross the BBB. MW is an important parameter in determining the free diffusion of small molecules across the BBB as well as BRB” (Line 283-286).
“On the contrary, larger lipophilic molecules and hydrophilic molecules require ATP-dependent transports to cross the barrier, including receptor-mediated vesicular transport, non-receptor-mediated pinocytosis, transporters, and pumps” (Line 301-303).
Reviewer 2 Report
In this manuscript, Ilenia Martinelli et al provided a comprehensive review on “Brain and retinal organoids for disease modeling: the importance of in-vitro blood-brain and retinal barriers studies”. The manuscript is focused mainly on the recent advances in the 3D-structure brain and retinal organoids used as a beneficial model for neurological and ophthalmological diseases and a screening tool for blood-brain barrier (BBB) and blood-retinal barrier (BRB) penetration. The manuscript convincingly explains not only the progress in the developments of 3D-strucuture brain and retinal, organs-on-a-chip, but also various obstacles of the developments to be overcome. Recent advanced models, microfluidic BBB and BRB chip-models still remain challenging; the necessity of improvement in their properties and functions. The readers would like to know how these obstacles of organoids can be overcome, that is, exemplifications to improve them. These challenges might pave the future avenue to develop the functional and integral in-vitro BBB and BRB models. Please add the future avenue more in Discussion section, if possible.
Author Response
Reviewer 2 Reply
-The readers would like to know how these obstacles of organoids can be overcome, that is, exemplifications to improve them. These challenges might pave the future avenue to develop the functional and integral in-vitro BBB and BRB models. Please add the future avenue more in Discussion section, if possible.
We thank the reviewer for the suggestion. Thus, we added the following paragraph in the discussion (Lines 505-517): “Additionally, BRB- and BBB-on-a-chip have been developed as microfluidic cell culture devices to overcome the limitations of static in vitro models. Modifying the architecture of the device allows the recreation of the physiological environment in vivo while measuring barrier function. The assessment of barriers in organs-on-chips can be difficult, but they offer the opportunity of continuous, non-invasive sensing of barrier quality, which allows better investigation of central aspects of pathophysiology, biological processes, and progress of therapies that target barrier tissues [204 Arık YB, van der Helm MW, Odijk M, Segerink LI, Passier R, van den Berg A, van der Meer AD. Barriers-on-chips: Measurement of the barrier function of tissues in organs-on-chips. Bio-microfluidics. 2018 Jun 26;12(4):042218. DOI: 10.1063/1.5023041]. The development of more accurate, and sophisticated barriers-on-a-chip with the capacity to grow in vitro connected with appropriate vascular supplies and nerves, pave the way for the development of functional and integral in vitro BRB and BBB models and offer a promising avenue by enabling future research scientists to perform experiments on a realistic replica when testing the effectiveness of novel experimental therapies [Williams-Medina et al. ref. 168, Ragelle et al. ref. 174].